# Effects of Physiological and Pathological Urea Concentrations on Human Microvascular Endothelial Cells

**DOI:** 10.3390/ijms24010691

**Published:** 2022-12-30

**Authors:** Graziano Colombo, Alessandra Altomare, Emanuela Astori, Lucia Landoni, Maria Lisa Garavaglia, Ranieri Rossi, Daniela Giustarini, Maria Chiara Lionetti, Nicoletta Gagliano, Aldo Milzani, Isabella Dalle-Donne

**Affiliations:** 1Department of Biosciences (Department of Excellence 2018–2022), Università degli Studi di Milano, 20133 Milan, Italy; 2Department of Pharmaceutical Sciences, Università degli Studi di Milano, 20133 Milan, Italy; 3Department of Biotechnology, Chemistry and Pharmacy (Department of Excellence 2018–2022), University of Siena, 53100 Siena, Italy; 4Department of Biomedical Sciences for Health, Università degli Studi di Milano, 20133 Milan, Italy

**Keywords:** urea, HMEC-1, CVD, CKD, vasorin, differential proteomics, EndMT

## Abstract

Urea is the uremic toxin accumulating with the highest concentration in the plasma of chronic kidney disease (CKD) patients, not being completely cleared by dialysis. Urea accumulation is reported to exert direct and indirect side effects on the gastrointestinal tract, kidneys, adipocytes, and cardiovascular system (CVS), although its pathogenicity is still questioned since studies evaluating its side effects lack homogeneity. Here, we investigated the effects of physiological and pathological urea concentrations on a human endothelial cell line from the microcirculation (Human Microvascular Endothelial Cells-1, HMEC-1). Urea (5 g/L) caused a reduction in the proliferation rate after 72 h of exposure and appeared to be a potential endothelial-to-mesenchymal transition (EndMT) stimulus. Moreover, urea induced actin filament rearrangement, a significant increase in matrix metalloproteinases 2 (MMP-2) expression in the medium, and a significant up- or down-regulation of other EndMT biomarkers (keratin, fibrillin-2, and collagen IV), as highlighted by differential proteomic analysis. Among proteins whose expression was found to be significantly dysregulated following exposure of HMEC-1 to urea, dimethylarginine dimethylaminohydrolase (DDAH) and vasorin turned out to be down-regulated. Both proteins have been directly linked to cardiovascular diseases (CVD) by in vitro and in vivo studies. Future experiments will be needed to deepen their role and investigate the signaling pathways in which they are involved to clarify the possible link between CKD and CVD.

## 1. Introduction

Uremic toxins are biologically active molecules with side effects on several physiological functions. Uremic toxins normally are excreted by the kidneys, while in chronic kidney disease (CKD) patients, they accumulate in the blood [1]. This retention phenomenon is dependent on the reduced renal function occurring in CKD, and it worsens with the progression of the pathology [2]. Urea is a water-soluble molecule produced during the urea cycle in the hepatocytes. This cycle is necessary to convert toxic ammonia, which originates from protein catabolism, into urea, subsequently excreted through the urine [3]. 

Urea is the uremic toxin with the highest plasma concentration [1,4], showing direct and indirect side effects on different organs and tissues [5]. Urea is reported to exert toxic effects on the gastrointestinal tract (contributing to epithelial barrier breakdown and microbiome alteration), on kidneys (indirectly promoting renal fibrosis), on adipocytes (inducing insulin resistance), on blood components (causing erythropoietin carbamylation), and on the cardiovascular system (CVS).

In this context, CKD patients are at high risk for cardiovascular disease (CVD) [6], as they show a high level of carbamylated low-density and high-density lipoproteins (cLDLs, cHDLs), proven to be a consequence of uremia [7] and leading to detrimental CV effects [8,9,10]. Furthermore, many studies reported that high level of blood urea nitrogen, a common uremic toxin, is associated with increased mortality and hospitalization in patients with CVD [11]. 

Notwithstanding that in vivo evidence suggests urea is a risk factor for mortality and atherosclerosis, the interpretation of clinical trial results is quite challenging, as patients with CKD have multiple comorbidities and urea acts together with other uremic toxins. In terms of mechanism, exposure to urea has been shown to induce reactive oxygen species (ROS) production in human aortic endothelial cells, leading to the activation of pro-inflammatory pathways and the inactivation of the anti-atherosclerosis enzyme PGI2 synthase [7] and in human arterial endothelial cells, causing alteration in mitochondrial proteins and in the expression of inflammatory markers [8]. Besides, in human aortic smooth muscle cells, urea affects BAD [B-cell lymphoma 2 (BCL2)-associated death promoter] expression, a pro-apoptotic member of the BCL2 family [9]. This phenomenon could contribute to the increased apoptosis observed in the arterial wall of patients with CKD and could promote vascular medial calcification [10].

Even though several papers support the emerging notion that urea is a direct and indirect uremic toxin in CKD, especially regarding the elevated CVD risk in these patients, the mechanisms of urea’s direct toxicity still require further investigation. In particular, the literature lacks studies exploring the effects of urea on protein expression and modification. In this study, we investigated some urea side effects potentially involved in CVD when vascular cells are exposed to physio-pathological urea concentrations found in healthy people or patients with CKD. We used a cell line from the microvasculature, as microcirculation is the principal seat of exchanges between circulation and tissues.

The endothelial cell (EC) monolayer is located at the interface between the blood and tissues, and it is involved in the regulation of fundamental functions such as vascular tone, permeability, and leucocyte trafficking. Its regulatory role requires, in many situations, long-distance communication. This function is accomplished by different strategies: (i) secretory pathways (such as cytokines, chemokines, complement proteins, coagulation, or growth factors), (ii) exocytosis of Weibel Palade Bodies, (iii) secretory granules, and (iv) extracellular vesicles (EVs) [12]. ECs constitutively secrete EVs into the blood under physiological conditions. Plasma levels of endothelial EVs have been found to be increased in patients with vascular diseases or in conditions involving EC injury or dysfunction [13]. Under pathologic or stress conditions, exosome secretion may increase, and/or exosomal content may change [14]. The content of EVs usually reflects the status of the donor cell and can influence the behavior of recipient cells both locally and systemically [15]. Chronic exposition of EC monolayer to urea could presumably alter the release rate and the composition of EVs. The investigation of secretome, using a large-scale proteomic approach, could unveil possible protein signaling molecules responsible for the long-distance effect induced by EC after exposition to the most abundant uremic toxin.

## 2. Results

### 2.1. Effect of Urea on the Growth Rate of HMEC-1

The growth of cultured HMEC-1 was followed up to 72 h by using the SRB assay (Figure 1). We observed that only the exposure of HMEC-1 for 72 h to the highest urea concentration tested (5 g/L), which is found in humans in pathological conditions, caused a significant reduction in the cell proliferation rate (*p* < 0.05). Control HMEC-1 and cells treated with urea 0.25 and 2 g/L grew exponentially over three days, whereas cells treated with urea 5 g/L showed a progressive reduction in growth over time. 

### 2.2. Urea-Induced Oxidative Stress Only in HMEC-1

Uraemia is related to oxidative stress [16]. Indeed, uremic toxins seem to contribute to CVD onset and progression in CKD, exacerbating oxidative stress and inflammation, which are non-traditional risk factors for CVD [17]. We evaluated urea-induced oxidative stress using protein carbonylation and oxidation of protein thiols as biomarkers. We did not find any difference in protein carbonylation as a result of the different times of exposure of the cells to the various concentrations of urea (data not shown). We measured a significant decrease in the total amount of protein thiols only in HMEC-1 treated with 2 g/L urea for 24 h (Figure 2). It is known that oxidative stress leads to the formation of unwanted disulphide bonds in the cytoplasm, causing a lowering in the total amount of thiols, eventually leading to impaired protein function. Nevertheless, thiol oxidation is reversible, and the results seem to indicate that the oxidation of protein thiols induced by urea and found at 24 h is subsequently recovered. 

### 2.3. Effect of Urea on Microfilaments Organization in HMEC-1

Since the rearrangement of both microtubules and microfilaments can contribute to endothelial dysfunctions [18,19], we evaluated tubulin and actin expression in HMEC-1 treated with urea. Based on the results of the Western blot, tubulin expression did not change throughout the duration of treatment at the various concentrations of urea while the amount of actin showed a tendency to decrease with increasing urea concentration, but the differences were not statistically significant (Figure 3). This observation was corroborated by immunofluorescence results. Comparing control cells with those exposed to 5 g/L urea for 72 h, actin signal in treated cells is weaker (Figure 4). In addition, in control cells, microfilaments are organized in randomized arrays, while in HMEC-1 exposed to 5 g/L urea for 72 h, they form peripheral bands (Figure 5).

A cytoskeleton modification is often accompanied by a modification in junctional proteins, all contributing to altering endothelial barrier permeability [20]. However, we found no significant differences in the expression of either VE-cadherin or beta-catenin.

### 2.4. Urea Induces Endothelial-to-Mesenchymal Transition in HMEC-1

Endothelial-to-mesenchymal transition (EndMT) is implicated in the pathogenesis of several CVD [21]. Remarkable morphological rearrangement of microfilaments is a feature of EndMT [22]; therefore, we measured the amount of another biomarker of EndMT, matrix metalloproteinase-2 (MMP-2), through zymography [23]. After 72 h of exposure to urea, HMEC-1 increased MMP-2 release in the medium in a concentration-dependent way (Figure 6). Results show a statistically significant difference between control cells and cells exposed to 5 g/L urea for 72 h (*p* < 0.05) and between cells exposed to 0.25 g/L urea and those exposed to 5 g/L urea for 72 h (*p* < 0.05). Thus, urea may represent a stimulus for EndMT, which in turn could play a role in CVD initiation and/or progression. 

### 2.5. Effect of Urea on Protein Expression in HMEC-1

For proteomic analysis, we compared the HMEC-1 treated with 0.25 g/L urea, the mean of the physiological urea concentration measurable in healthy subjects, with those treated with 5 g/L urea, a concentration measurable in CKD patients. We did not consider the control cells because this condition, i.e., the absence of urea, does not exist physiologically, even in healthy individuals. We analyzed by proteomics all the conditions, including the control. Having to choose which comparison to show and which to dwell on, it seemed more logical to compare the physiological condition (i.e., 0.25 g/L urea) with the pathological one. All the other experiments performed allowed us to compare (by ANOVA analysis or qualitatively) the various conditions between them, even in pairs. Proteomic analysis does not allow us to make a simultaneous multiple comparison between different conditions to extrapolate a picture of variations, but only to compare one condition against another. We have chosen the closest to the human physiological condition to hypothetically obtain results as realistic as possible. Table 1 and the volcano plot (Figure 7) show that only a few proteins resulted in being up- or down-regulated when comparing cells treated with 0.25 g/L urea vs. 5 g/L urea. To check if these proteins were connected to each other according to their functions, we performed an analysis with STRING obtaining the network of up-regulated proteins (Figure 8) and the network of down-regulated proteins (Figure 9). The two networks do not reveal significantly more interactions than expected, according to STRING lambda calculation. This means that our sets of proteins are composed of an apparently random collection of proteins that are poorly connected to each other or whose interactions are not yet known by STRING based on available data.

### 2.6. Effect of Urea on the HMEC-1 Secretome

We performed a secretome analysis of HMEC-1 exposed to urea, comparing control cells with those treated with 5 g/L urea, a concentration measurable in CKD patients. Table 2 and the volcano plot (not shown) showed that only a few proteins secreted in the medium resulted in being up- or down-regulated when comparing control cells with those treated with 5 g/L urea. To check if these proteins were connected to each other according to their functions, we performed an analysis with STRING obtaining the network of up-regulated proteins (Figure 10) and the network of down-regulated proteins (Figure 11). The two networks do not reveal significantly more interactions than expected, according to STRING lambda calculation. This means that our sets of proteins are composed of an apparently random collection of proteins that are poorly connected or whose interactions are not yet known by STRING-based available data.

### 2.7. Effect of Urea on Vasorin Expression in HMEC-1

Within the set of secretome proteins significantly regulated by urea, we focused particularly on vasorin, given its involvement in cellular mechanisms closely linked to CKD and CVD. Vasorin is a glycoprotein existing in three forms: a cell-surface, transmembrane form [24], an extracellular (secreted) form [24,25,26] whose signaling pathway is linked to the transforming growth factor-beta (TGF-ß)-mediated epithelial-to-mesenchymal (EMT) transition, and an intracellular form [27]. Moreover, it has been recently proposed as a potential biomarker for nephropathies and tumorigenesis [28], and it is involved in acute arterial injury and in age-associated vascular remodeling [29]. In addition to having these detrimental effects, vasorin protects vascular smooth muscle cells against calcification and osteo-/chondrogenic trans-differentiation [30]. Overall, vasorin appears to be connected with both kidney pathologies and CVD.

Our proteomic analysis of the HMEC-1 secretome, which compared control cells with those exposed to 5 g/L urea for 72 h, showed that the secreted form of vasorin was downregulated with a nine-fold decrease. The Western blot results confirmed a significant decrease (about a 188-fold decrease) in the expression of the secreted form of vasorin in the medium of HMEC-1 treated with 5 g/L urea for 72 h, compared to control cells (Figure 12).

## 3. Discussion

CKD is a non-communicable disease with a world prevalence of 8–16% [6]. It is diagnosed when there is a decreased kidney function shown by glomerular filtration rate (GFR) less than 60 mL/min (established for a reference man with 1.73 m² body surface area), or markers of kidney damage, or both, of at least three months duration [31]. Complications and morbidities such as CVD, acute kidney injury, bone disorder, mineral balance disorder, hospitalization, anemia, oxidative stress, chronic inflammation, and dysbiosis increase in parallel with the GFR decline [2]. The decreased kidney function is responsible for the retention of several molecules, which normally are in part eliminated through feces and in part cleared by kidneys, so they accumulate in CKD and are called uremic toxins [32]. Interest in uremic toxins is continually enhancing since they have been recognized as causes of a lot of side effects [33,34]. 

Urea seems to exert toxic effects on the gut, kidneys, adipocytes, blood components, and CVS [5]. High levels of urea have been correlated with a higher rate of mortality and hospitalization in CKD patients with CVD [11] and with a higher level of cLDLs, which represent a pro-atherogenic factor [35]. Therefore, urea could contribute, at least in part, to the increased risk of CVD observed in patients with CKD compared to the normal population [6]. However, the interpretation of clinical trial results is complex since CKD patients have multiple comorbidities, and urea acts together with the other uremic toxins. Therefore, in this study, we tried to determine the effects of urea that potentially link it to CVD, utilizing physio-pathological concentrations of this toxin. We tested them on an endothelial cell line from the microvasculature since, in the literature, there are no studies that have evaluated urea effects on microcirculation, even though it is the principal set of exchanges between circulation and tissues. We tested the following concentrations: 0.25, 2, and 5 g/L, after consulting the European Uremic Solutes Database (EUTox-DB), which reports the following concentrations: 0.30 (+/−0.12) (0.30–0.40) g/L grand mean in healthy subjects; 1.57 (+/−0.64) (1.30–4.60) g/L grand mean in uremic patients.

Urea affects cell proliferation. We observed that the exposure of HMEC-1 for 72 h to the highest concentration of urea tested (5 g/L) caused a significant decrease in the cell proliferation rate (*p* < 0.05). This could depend on a pro-apoptotic effect of urea. An increase in BAD [B-cell lymphoma 2 (BCL2)-associated death promoter] expression, a pro-apoptotic member of the BCL2 family, was measured in human aortic smooth muscle cells exposed to 20 mM (equal to 1,2 g/L) urea [15]. Apoptosis of the arterial cells is a key event in vascular remodeling and in the progression of atherosclerosis, and CKD patients show increased apoptosis in their arterial wall [16]. So, the pro-apoptotic effect of urea could contribute to medial vascular calcification.

Non-traditional uremic risk factors for vascular dysfunction and calcification include chronic oxidative stress and inflammation [36,37]. Patients with CKD show a high level of oxidative stress, to which uremic toxins also contribute [38]. Incubation with 20 mM urea resulted in an increased production of reactive oxygen species (ROS) in human aortic endothelial cells [13], human arterial endothelial cells [14], and 3T3-L1 adipocytes [39]. After 24, 48, and 72 h of HMEC-1 exposure to urea, we measured protein carbonylation and the total amount of protein thiols [40], whose concentration, respectively, increases and decreases in the presence of oxidative stress. However, we did not find any significant difference in protein carbonylation and found only a trend toward reduction in the total amount of protein thiols in cells treated for 24 h with pathological concentrations of urea (2–5 g/L), with a significant difference compared to control cells only in cells treated with 2 g/L urea. Since thiol oxidation is a mainly reversible modification, we could hypothesize that the initial oxidation of protein thiols induced by urea was later recovered by some intracellular antioxidant system.

Another emerging risk factor for CVD is the EndMT, a complex biological process during which endothelial cells start to express mesenchymal cell-specific proteins and progressively reduce the expression of endothelial cell-specific proteins [41,42]. Therefore, endothelial cells undergo a transition towards a more mesenchymal-like phenotype, mainly induced by the members of the TGF-β family. EndMT is implicated in the pathogenesis of several diseases, such as malignant diseases, fibrotic diseases, pulmonary arterial hypertension, atherosclerosis, diabetes mellitus, cavernous malformation, and fibrodysplasia ossificans progressiva [22]. To investigate a possible EndMT in HMEC-1 exposed to urea, we evaluated cell morphology and cytoskeleton organization, the expression of some junctional proteins, and the total amount of MMP-2 secreted by the cells. 

We measured a slight reduction, although not statistically significant, in actin expression in HMEC-1 exposed to the pathological concentrations of urea. In addition, after a 72-h exposure to 5 g/L urea, actin microfilaments formed peripheral bands rather than organizing themselves in randomized arrays. Urea, therefore, induces a morphological rearrangement of microfilaments, which is also a feature of EndMT [22].

A cytoskeleton modification is often accompanied by modifications in junctional proteins, all contributing to altering endothelial barrier permeability [20]. In addition, junctional proteins, such as VE-cadherin, are among those proteins whose expression decreases or disappears during EndMT, which is characterized by loss of cell-cell junctions and polarity [43]. However, we found no significant differences in VE-cadherin or beta-catenin expression. Other EndMT features are the acquisition of cellular motility and invasive properties. In this regard, we measured the amount of secreted MMP-2 through zymography. MMPs degrade extracellular matrix proteins, playing important roles in vascular tissue remodeling processes. They can influence cell migration, proliferation, contraction, and calcium signaling [44]. MMP alterations are correlated with CVD, such as hypertension, atherosclerosis, excessive venous dilation, and lower extremity venous disease [45]. A 72-h exposure of HMEC-1 to various concentrations of urea increased MMP-2 release in the culture medium in a concentration-dependent way. Thus, urea may represent a stimulus for EndMT, which in turn could play a role in CVD initiation and/or progression. 

A previous study compared protein expression in HUVECs treated with uremic serum, i.e., serum from patients with CKD undergoing hemodialysis or with normal serum taken from healthy subjects. Results mainly showed differential expression in proteins linked to inflammation, oxidative stress, and the cytoskeleton [46]. Uremic serum contains all the uremic toxins. In the present study, we performed proteomic analyses to study the differential protein expression induced by a single uremic toxin, urea, in both the intracellular proteome and the secretome of HMEC-1, with the aim of better clarifying the effects due specifically to this molecule. The volcano plots show that only a few proteins, both intracellular and secreted, were up- or down-regulated following the HMEC-1 exposure for 72 h to the highest urea concentration (5 g/L). The subsequent analysis of the proteomics data with the software STRING, gathering proteins according to their functions, did not reveal metabolic pathways markedly influenced by cell exposure to urea, either in relation to the intracellular proteome or in relation to the secretome. However, some of the secreted proteins, whose expression turned out to be significantly up- or down-regulated by exposure of HMEC-1 to 5 g/L urea, are particularly interesting and deserve further investigation.

One of the aims of our study was to evaluate the change in the secretome of ECs exposed to a pathological concentration of urea in order to identify possible soluble markers specifically released following cell exposure to urea. In particular, we aimed to identify proteins that could be used by cells as short- or long-distance signaling molecules and that could also be involved in the mechanisms that lead to CVD. The results of the proteomic analysis revealed a long list of candidate proteins that can be considered reliable. 

Annexin A5 and programmed cell death 5 are significantly over-expressed in the secretome after exposure to urea, corroborating the hypothesis that urea could have pro-apoptotic effects [15]. In fact, annexin A5 binds with high affinity to negatively charged phospholipids such as phosphatidylserine (PS), which is translocated from the inner layer of the plasma membrane to the outer layer of apoptotic cells as a signal for phagocytes [47]. Programmed cell death 5 overexpression cannot directly induce cell apoptosis, but it can enhance it [48]. Apoptosis plays an important role in the loss of cells during myocardial infarction and heart failure [49], and it is strongly correlated with the development of atherosclerotic plaque vulnerability [50]. Annexin A5 has been proposed as an imaging biomarker of CV risk since preclinical and clinical studies showed that exposure of PS on the cell surface in the CVS is an attractive biological target in atherosclerosis, heart failure, and cardiac ischemia [51]. The urea pro-apoptotic effects deserve further investigation, which could reveal the link between urea and CVD.

Keratin, fibrillin-2, and collagen IV are other proteins differentially secreted in the secretome, and they support the hypothesis that urea can be a stimulus for EndMT. Urea-induced down-regulation of keratin and a loss of keratin is a hallmark of EMT [52]. The alteration or the disruption of the keratin cytoskeleton makes epithelia susceptible to tissue damage and various stresses [52], and it leads to an increased migration of cancer cells [53]. Fibrillin-2 and collagen IV were up-regulated by HMEC-1 exposure to urea. Fibrillin-2 is a cysteine-rich glycoprotein that supports structures essential to maintain tissue integrity and which regulates signaling events [54]; it is employed as a marker of EMT induced by TGF-ß [55]. Collagen IV is a principal component of epithelial basement membranes, where it organizes in sheet-like networks. It is produced by mesenchymal cells and by cancer cells that have undergone EMT [56]. It is an EMT biomarker since it contributes to the microenvironment remodeling typical of cells after EMT [57]. These results corroborate the hypothesis that urea could induce EndMT, which in turn could link urea and CVD [21].

The two most interesting secreted proteins undergoing downregulation are dimethylarginine dimethylaminohydrolase (DDAH) and vasorin. Both proteins have been directly linked to CVD by in vitro and in vivo studies.

DDAH is the enzyme responsible for the degradation of asymmetric dimethylarginine (ADMA), which is an important regulator of nitric oxide production. A downregulation of DDAH leads to a high level of ADMA, which is considered a CVD risk factor [58]. DDAH downregulation or knockdown causes endothelial dysfunction, increases systemic vascular resistance, and elevates systemic and pulmonary blood pressure [59]. Interestingly, ADMA is not only a CVD risk factor but also a CKD progression risk factor, making DDAH an important regulator of renal and vascular function integrity. Enhancing DDAH activity has been proposed as a therapeutic strategy to prevent CVD and CKD progression [60]. Patients with CKD show high levels of ADMA [61], explained by higher levels of protein methylation; increased rate of protein turnover; impaired activity of DDAH; impaired renal excretion [62]. Based on the results of our study, we can hypothesize another cause to add to this list, namely higher levels of urea, since HMEC-1 exposure to urea induces a marked downregulation of DDAH, which in turn will give rise to an increase in ADMA level. Further studies are needed to confirm this hypothesis, which could link urea to CVD and CKD progression. 

The choice to validate vasorin by Western blot derives from the available literature; vasorin represents a good and reasonable candidate soluble marker, specifically released after cell exposure to urea, due to different important features:(1)Vasorin is abundantly overexpressed after cell exposure to urea. This evident modulation reduces the possibility of having false positive proteins.(2)Intramembrane vasorin can be cleaved by ADAM17, a disintegrin, and metalloprotease 17, releasing the extracellular portion. This soluble and active protein binds to TGF-β and prevents its interaction with its specific receptor [26]. The solubility of released vasorin allowed us to easily verify its abundance also in plasma samples of CKD patients compared to control subjects, confirming the modulation of vasorin in human plasma.(3)Vasorin is a cell surface single-pass transmembrane glycoprotein, and it has been shown to be abundantly expressed also by vascular smooth muscle cells. Therefore, soluble vasorin could be a candidate able to cooperate and contribute to the crosstalk between endothelial cells and vascular smooth muscle cells at the systemic level [24,28].(4)Vasorin has also been demonstrated to bind TGF-β, blocking its biological activity [25]. It also regulates Notch1 signaling by interacting with Numb and preventing the degradation of Notch1 [63]. These two signaling pathways are reported to be important in the homeostasis of vascular smooth muscle cells [25,63], and vasorin could be an important signaling mediator in the crosstalk between vascular smooth muscle cells and the endothelial cell monolayer. In addition, vasorin expression is altered in several diseases; it is higher than normal in synovial fluid of patients with osteoarthritis [64], in plasma of patients with diabetic nephropathy [65], in serum of subjects with hepatocarcinoma [66], in urinary exosomes of patients with thin basement membrane nephropathy; on the contrary, vasorin expression is lower than normal in subjects with early IgA nephropathy [67,68]. Our proteomic analysis of the HMEC-1 secretome was confirmed by Western blot analysis. Vasorin downregulation was previously observed also in vivo after vascular injury. As a consequence, the expression of several cytokines, including TGF-ß, was upregulated, leading to neointimal formation, the typical fibroproliferative response to vascular injury [26]. Reverting vasorin downregulation significantly diminished injury-induced vascular lesion formation. For these reasons, vasorin has been proposed as a potential therapeutic target for vascular fibroproliferative disorders [24]. Vasorin is also linked to EMT. Vasorin is effectively cleaved by activated MMP-2 both in vitro and in vivo. We measured a significant increase in the amount of MMP-2 secreted by HMEC-1 exposed to urea. Therefore, we can hypothesize that the increased MMP-2 may have contributed to increasing the release of the soluble form of vasorin. Extracellular vasorin interferes with TGF-ß–mediated EMT, modulating E-cadherin expression and actin filament organization [25], as we have also observed in HMEC-1. In addition, vasorin modulates collagen expression [24], another EMT marker that we found among the differentially expressed proteins in HMEC-1 exposed to urea. So, vasorin downregulation, promoting TGF- ß pathway, could represent an EndMT-inducing stimulus. Since ADAM17 is the principal metalloprotease controlling vasorin cleavage, it would be interesting to investigate whether ADAM17 modulation can prevent EndMT.

We can conclude that urea at pathological concentrations affects cell proliferation and microfilament organization and induces EndMT in HMEC-1. Proteomic analysis also confirms a dysregulation in the expression of proteins involved in pro-apoptotic pathways and EndMT, in addition to the alteration of some proteins directly linked to CVD. The modest effects of urea observed in our study, sometimes in contrast to what is reported in the literature, could be explained by using only urea concentrations found in pathophysiological conditions in humans, i.e., in healthy individuals and those with CKD. Moreover, we added urea to the cell medium on the first day of the various experiments and evaluated its effects after 24, 48, or 72 h. So, it is possible that over time the cells may metabolize urea, or urea could degrade in the culture medium and, therefore, reduce in concentration over time, thus showing limited effects at long times. However, despite these limitations, urea at concentrations found in CKD reduced cell proliferation and induced reorganization of microfilaments and EndMT in cultured HMEC-1.

Further studies are needed to better define the toxic effects of urea. Our proteomics results may be the starting point for evaluating the variation in expression of proteins most related to CVD and which we have shown to be modulated by urea in other cell lines and in vivo in the plasma sample of healthy subjects and patients with CKD.

## 4. Materials and Methods

### 4.1. Cell Culture

The human dermal microvascular endothelial cells-1 (HMEC-1) were obtained from the American Type Culture Collection (Manassas, VA, USA) and grown in plates with MCDB 131 Medium (Sigma-Aldrich, Milan, Italy), supplemented with 10% fetal bovine serum (FBS, Euroclone, Milan, Italy), 2 mM L-glutamine, 100 U/mL penicillin, 100 μg/mL streptomycin, 10 ng/mL epidermal growth factor, 0.1 ug/mL hydrocortisone (Sigma-Aldrich, Milan, Italy). Cell cultures were maintained at 37 °C with 5% CO_2_ and passaged every 3–4 days. For experiments, HMEC-1 cells were cultured in the presence or absence of different concentrations of urea (Sigma-Aldrich, Milan, Italy) for 24, 48, or 72 h.

### 4.2. Treatment of HMEC-1 with Urea

HMEC-1 were seeded at a concentration of 15,000 cells/cm^2^ and grown for 24 h, at 37 °C, with 5% CO_2_. Then, half of the medium was removed and replaced with an equal volume of solution with or without urea. This expedient is necessary since HMEC-1 release growth factors in the medium, and a complete replacement of the culture medium would slow down cell growth. Urea solutions were prepared by dissolving urea powder in phosphate buffered saline (PBS), obtaining a mother solution with a concentration of 100 g/L, which was diluted in complete medium (prepared as described before) at the following concentrations: 0.5, 4, and 10 g/L. These concentrations are double the desired ones because only half of the medium was changed. Urea solutions were filtered through a syringe with a 0.22-μm pore-sized filter to remove bacteria and particulate and added to the cell cultures at the final concentrations: 0.25, 2, and 5 g/L (equal to, respectively, 4, 33, and 83 mM). Control cells were treated in a similar way treatment, without the uremic toxin. In this way, all the treatment solutions contained the same volume of PBS, and they differed only for the presence or absence of urea. The treatment lasted 24, 48, or 72 h without changing the medium.

### 4.3. Proliferation Assay

Sulforhodamine B (SRB) assay is a colorimetric test that allows for quantifying cellular protein content, and it is largely used to indirectly quantify cell proliferation [69]. Briefly, cells were seeded and treated as described before in 24-multiwell plates. At each time point, cells were fixed with 50% trichloroacetic acid (Sigma-Aldrich, Milan, Italy, cod. T6399) for 2 h at 4 °C, then washed five times with Milli-Q water. Then, 0.04% (*w*/*v*) SRB dye (Sigma-Aldrich, Milan, Italy, cod. S1402), dissolved in 1% acetic acid, was added to each well and incubated at room temperature for 30 min; then, each well was washed four times with 1% (*v*/*v*) acetic acid and left to air-dry at room temperature. Finally, 1.2 mL of 10 mM Tris base solution (pH 10.5) was added to each well, and the plate was shaken on an orbital shaker for 10 min to solubilize the protein-bound dye. The absorbance at 490 nm was detected using a multimode microplate reader (EnSight Multimode Plate Reader, PerkinElmer, Waltham, MA, USA).

### 4.4. Quantification of Proteins Thiols 

Cell protein extracts were obtained by lysing cells with ice-cold lysis buffer [50 mM Tris-HCl, pH 7.4, 150 mM NaCl, 1% TRITON X-100, 0.1% SDS, 0.5% sodium deoxycholate supplemented with protease inhibitors (Sigma-Aldrich, Milan, Italy, P8340)]. Each lysate was incubated on ice for 30 min and centrifuged at 10,000× *g* for 10 min at 4 °C to remove cell debris. Protein concentration in supernatants was determined by bicinchoninic acid (BCA) protein assay. Protein thiol groups were measured by a biotin-maleimide assay. Briefly, 40 mM biotin-maleimide stock solution was prepared in dimethyl sulphoxide and stored at −20 °C. Then, 1 mg/mL of protein was incubated with 75 μM biotin-maleimide solution for 1 h at room temperature and mixed with Laemmli sample buffer (2% SDS, 20% glycerol, 125 mM Tris-HCl, pH 6.8), heated for 5 min at 90 °C and separated on 12% SDS-PAGE stain-free gel (Bio-Rad Laboratories, Segrate, Italy) [70]. Separated proteins were electroblotted onto a low-fluorescence polyvinylidene difluoride (LF-PVDF) membrane. LF-PVDF membrane was washed with PBST [10 mM Na-phosphate, pH 7.2, 0.9% (*w*/*v*) NaCl, 0.1% (*v*/*v*) Tween-20 (Sigma-Aldrich, Milan, Italy, cod. P9416)] [70] and blocked for 1 h in 5% (*w*/*v*) non-fat dry milk in PBST. After washing three times with PBST for 5 min, the biotin tag was probed by a 2-h incubation with 5% non-fat dry milk/PBST containing streptavidin-HRP (1:5000 dilution, Euroclone, Milan, Italy). Biotinylated proteins were visualized by enhanced chemiluminescence (ECL) detection (Bio-Rad Laboratories, Segrate, Italy, cod. 1705061) using the ChemiDoc Touch Imaging System (Bio-Rad Laboratories, Segrate, Italy). ECL signals were normalized on LF-PVDF stain-free signals [71].

### 4.5. Western Blot 

Proteins from cell extracts were separated and transferred to LF-PVDF membrane as described previously. After washing three times with PBST for 5 min, the membrane was incubated for 2 h with 5% non-fat dry milk/PBST containing the following primary antibodies: anti-actin (1:2000, Abcam, Cambridge, UK); anti-tubulin (1:40,000, Abcam, Cambridge, UK). The membrane was washed three times with PBST for 5 min and then incubated for 1 h with the following secondary antibodies, respectively: anti-mouse (1:10,000) and anti-rabbit (1:20,000). Proteins of interest were visualized by ECL detection (Bio-Rad Laboratories, Segrate, Italy, cod. 1705061) using the ChemiDoc Touch Imaging System (Bio-Rad Laboratories, Segrate, Italy). ECL signals were normalized on LF-PVDF stain-free signals [71] using tubulin as a housekeeping protein.

### 4.6. Immunofluorescence

HMEC-1 were cultured on 12-mm diameter round coverslips, seeded at a concentration of 15,000 cells/cm^2^ on 24-well culture plates, and treated with urea as described before. At each time point, cells were washed in PBS, fixed in 4% paraformaldehyde in PBS containing 2% sucrose for 10 min at room temperature, post-fixed in 70% ethanol, and stored at −20 °C until use. For microtubule analysis, cells were washed in PBS three times, incubated for 5 min at room temperature with 0.1% Triton X-100/PBS, and blocked with 1% bovine serum albumin (BSA) in PBS for 1 h. Cells were then incubated with the primary monoclonal anti-tubulin antibody (1:300, diluted in 0.5% BSA/PBS, Abcam, Cambridge, UK) at 4 °C overnight. The next day cells were washed four times with PBS and incubated for 1 h in the dark with the secondary antibody, an anti-rabbit tetramethylrhodamine-isothiocyanate (TRITC)-conjugated (Abcam, Cambridge, UK) diluted 1:200 in 0.5% BSA/PBS (Abcam, Cambridge, UK), and washed extensively in PBS. For microfilament detection, cells blocked with 1% BSA were incubated for 1 h in the dark with fluorescein isothiocyanate (FITC)-phalloidin 1:1000 (Abcam, Cambridge, UK) in 1% BSA/PBS (Abcam, Cambridge, UK). After the labeling procedure was completed, the coverslips were incubated for 10 min with 4’6-diamidino-2-phenylindole (DAPI) and mounted onto glass slides using Mowiol mounting medium. Fixed cells were imaged with a ViCo confocal microscope (Nikon Europe B.V., VX Amstelveen, The Netherlands) and TCS NT confocal laser scanning microscope (Leica Microsystems Srl, Buccinasco, Milan, Italy).

### 4.7. Zymography

ProMMP-2 protein levels were assessed in the supernatants of cultured HMEC-1 by SDS-zymography. HMEC-1 were seeded on 24-well culture plates at a concentration of 15,000 cells/cm^2^; after 24 h, cells were treated with urea as described above. Supernatants of cells exposed to urea for 24 or 72 h were collected and concentrated in an AmiconY10 at 6500× *g* for 15 min at 4 °C. The concentrated culture media were mixed 3:1 with sample buffer containing 10% SDS. Samples (4 μg of total protein) were run at 4 °C under non-reducing conditions and without heat denaturation onto 7.5% polyacrylamide gels (SDS-PAGE) co-polymerized with 1 mg/mL type I gelatin. After SDS-PAGE, the gels were washed twice in 2.5% Triton X-100 for 30 min each and incubated overnight in a substrate buffer at 37 °C (Tris–HCl 50 mM, CaCl_2_ 5 mM, NaN3 0.02%, pH 7.5). The gelatinolytic activity of matrix metalloproteinases (MMPs) was detected after staining the gels with Coomassie brilliant blue R250 as clear bands on a blue background and quantified by densitometric scanning (UVB and Eppendorf, Italy). To confirm the identity of MMP gelatinolytic activity, purified MMP-1 and MMP-2 (100 ng, Calbiochem, San Diego, CA, USA) were run as controls.

### 4.8. Quantitative Proteomic Analysis of HMEC-1 after a 72-h Exposure to Urea

HMEC-1 were seeded and exposed to urea for 72 h, as described above, without changing the medium. After the removal of the medium and three washes with PBS, cell protein extracts were obtained by lysing cells with the following lysis buffer: 8 M urea, 100 mM Tris-HCl, pH 8.5, and protease inhibitors (Sigma-Aldrich, Milan, Italy, P8340). Each lysate was incubated for 30 min at room temperature and centrifuged at 14,000× *g* for 30 min at 4 °C to remove cell debris. Protein concentration in supernatants was measured by BCA protein assay. To check the integrity of extracted proteins, part of the lysate was mixed with Laemmli sample buffer, heated for 5 min at 90 °C, and separated on 12% SDS-PAGE stain-free gel (Bio-Rad Laboratories, Segrate, Italy) [70]. Protein gel was acquired using the ChemiDoc Touch Imaging System (Bio-Rad Laboratories, Segrate, Italy). The rest of each lysate was used to perform tryptic digestion of proteins as described below.

Ten micrograms of proteins were mixed in 36 µL of 50 mM ammonium bicarbonate (AMBIC) dissolved in MS-grade water (Sigma-Aldrich, Milan, Italy). The pH was checked to ensure that it was around pH 8.0–8.5. Then, 5 mM dithiothreitol (DTT, diluted in AMBIC) was added, and samples were incubated in a Thermomixer at 600 rpm, 52 °C for 30 min. At this point, 15 mM iodoacetamide (IAM, diluted in AMBIC) was added, and samples were incubated in a Thermomixer at 600 rpm, at room temperature for 20 min, in the dark; 0.5 µg trypsin in 50 mM acetic acid was added (after activation for 15 min at 30 °C) respecting a ratio 1:20 trypsin:protein. Samples were incubated in a Thermomixer at 600 rpm, 37 °C overnight. The day after, 2 µL of 50% trifluoroacetic acid (TFA, diluted in MS-grade water) was added, and the pH was checked to ensure that it was lower than 2. 

### 4.9. Quantitative Proteomic Analysis of Proteins Released by HMEC-1 after a 72-h Exposure to Urea

HMEC-1 were seeded as described above. After 24 h, all the medium was removed and replaced with a medium without FBS, plus or minus urea. This expedient was necessary because the presence of serum in the medium was not compatible with proper protein separation using Bio-Gel P6 columns (Bio-Rad Laboratories, Segrate, Italy). HMEC-1 were exposed to urea for 72 h without changing the medium. Then, 30 mL of medium was collected for each condition and freeze-dried for 48 h. Each lyophilized sample was resuspended in 1.5 mL MS-grade water in order to increase the concentrations of proteins by 20 times. Samples were then processed using Bio-Gel P6 columns (Bio-Rad Laboratories, Segrate, Italy), according to the manufacturer’s instructions, to isolate proteins. An additional concentration (by a factor of 10) of the samples was obtained using a Savant SpeedVac Concentrator. The protein content of the concentrated fractions of the control and urea-treated medium was measured by BCA protein assay. To check the integrity of proteins, a small amount of each sample was mixed with Laemmli sample buffer, heated for 5 min at 90 °C, and separated on 12% SDS-PAGE stain-free gel (Bio-Rad Laboratories, Segrate, Italy) [70]. Protein gel was acquired using the ChemiDoc Touch Imaging System (Bio-Rad Laboratories, Segrate, Italy. The rest of each sample was digested as described above in Section 4.8.

### 4.10. High Resolution Mass Spectrometry Analysis (nLC-MSMS)

Tryptic peptides were analyzed at UNITECH OMICs (Università degli Studi di Milano, Italy) using a Dionex Ultimate 3000 nano-LC system (Sunnyvale, CA, USA) connected to an Orbitrap Fusion™ Tribrid™ Mass Spectrometer (Thermo Scientific, Bremen, Germany) equipped with a nano-electrospray ion source. Peptide mixtures were pre-concentrated onto an Acclaim PepMap 100−100 mm, 2 cm C18 and separated on an EASY-Spray column,15 cm, 75 mm ID packed with Thermo Scientific Acclaim PepMap RSLC C18, 3 mm, 100 Å. The temperature was set to 35°C, and the flow rate was 300 nL min^−1^. Mobile phases were the following: 0.1% formic acid (FA) in water (solvent A); 0.1% FA in water/acetonitrile (solvent B) with a 2/8 ratio. Peptides were eluted from the column with the following gradient: 4% to 28% of B for 90 min and then 28% to 40% of B in 10 min, and to 95% within the following 6 min to rinse the column. The column was re-equilibrated for 20 min. Total run time was 130 min. One blank was run between triplicates to prevent sample carryover. MS spectra were collected over an m/z range of 375–1500 Da at 120,000 resolutions, operating in the data-dependent mode, cycle time of 3 s between master scans. HCD was performed with collision energy set at 35 eV. Each sample was analyzed in three technical triplicates. LTQ raw data were searched against a protein database using SEQUEST algorithm in Proteome Discoverer software version 2.2 (Thermo Scientific, Bremen, Germany) for peptide/protein identification. The searches were performed against Uniprot KnowledgeBase (KB) (taxonomy *Homo sapiens*). The minimum peptide length was set to six amino acids, and enzymatic digestion with trypsin was selected, with a maximum of 2 missed cleavages. A precursor mass tolerance of 8 ppm and fragment mass tolerance of 0.02 Da were used; acetylation (N-term) and oxidation (M) were used as dynamic modifications, and carbamidomethylation (C) as a static modification. The false discovery rates (FDRs) at the protein and peptide level were set to 0.01 for highly confident peptide-spectrum matches and 0.05 for peptide-spectrum matches with moderate confidence. We considered only proteins with a score of coverage > 2% with at least two identified peptides. Differences in abundance ratio (AR) of proteins between control and treated samples were considered only with at least a 2-fold change and with a standard deviation between replicates less than 20%.

### 4.11. MS Data Analysis–Label-free Quantitative Proteomics 

Resulting MS raw files from all the technical and biological replicates were analyzed by using Proteome Discoverer software (Version 2.4.0.305) based on the SEQUEST algorithm as a database search engine. Database search against the latest Human UniProtKB/SwissProt FASTA files Release (UniProt release 2019_11-18 December 2019) was performed according to the following parameters: trypsin was specified as a proteolytic enzyme, cleaving after lysine and arginine except when followed by proline. Up to two missed cleavages were allowed. The precursor ion tolerance was set to 10 ppm, and the fragment tolerance was set to 0.6 Da. Carbamidomethylation of cysteine was defined as fixed modification, while oxidation of methionine and acetylation at the protein N-terminus were specified as variable modifications. The FDR for peptide identification was calculated using the Percolator algorithm in the Proteome Discoverer workflow based on the search results against a decoy database and was set at 1% FDR. Identified peptides were quantified by a typical Processing workflow for Minora feature detection, based on the quantification of isotopic clusters regardless of whether or not they are associated with a Peptide Spectral Match. The RT alignment was performed with a maximum RT shift of 10 min. For quantification, all unique and razor peptides were considered, and the normalization of intensity values was performed over precursor (consensus features) against the total peptide amount. Samples were previously categorized by the cell line (cell-line 1–21) and by the treatment type (control–urea 5 g/L, urea 0.25 g/L–urea 5 g/L), and for the identification of differentially regulated proteins, quantification jobs were alternatively launched using the individual ratios option. Ratio calculation was based on Pairwise Ratio based approach using summed abundances for single proteins abundance calculations. Proteins were grouped applying strict parsimony principle and filtered at a 1% FDR at the protein level and further categorized based on annotation aspects (Biological Process, Molecular Function, and Cellular Components). 

### 4.12. Validation of Proteomics Data of Proteins Released by HMEC-1 after a 72-h Exposure to Urea

A small amount of the concentrated fractions of control and urea-treated medium (obtained as described above in Section 4.9.) was mixed with Laemmli sample buffer, heated for 5 min at 90 °C, and separated on 12% SDS-PAGE stain-free gel (Bio-Rad, Laboratories, Segrate, Italy) [70]. After washing three times with PBST for 5 min, the membrane was incubated overnight with 5% non-fat dry milk/PBST containing the primary antibody: anti-vasorin (1:5000, Abcam, Cambridge, UK). The membrane was washed three times with PBST for 5 min and then incubated for 1 h with an HRP-conjugated goat anti-rabbit secondary antibody (1:5000). Vasorin was visualized by ECL detection (cod. 1705061, Bio-Rad Laboratories, Segrate, Italy) using the ChemiDoc Touch Imaging System (Bio-Rad Laboratories, Segrate, Italy).

## Figures and Tables

**Figure 1 ijms-24-00691-f001:**
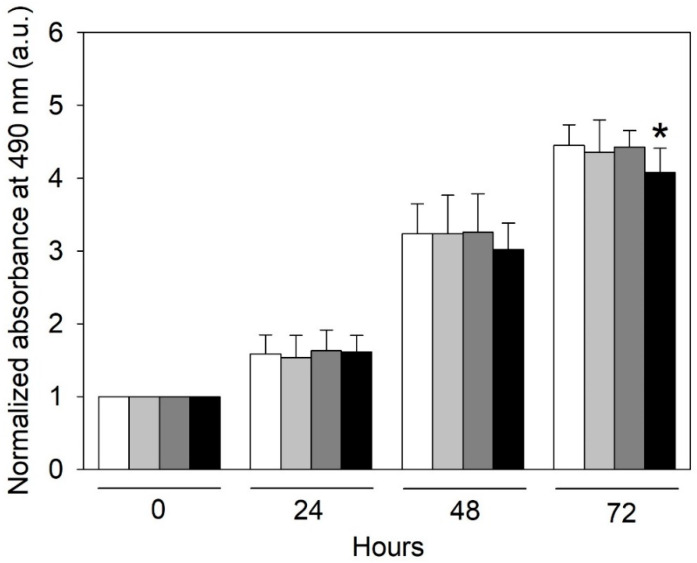
Effect of urea on HMEC-1 proliferation measured indirectly by the SRB assay. Histograms showing the mean absorbance measured at 490 nm in control cells (white histogram) and cells treated with 0.25 (light gray histogram), 2 (dark gray histogram), or 5 (black histogram) g/L urea for 0, 24, 48, or 72 h. Data are expressed as the mean ± SD. * *p* < 0 05.

**Figure 2 ijms-24-00691-f002:**
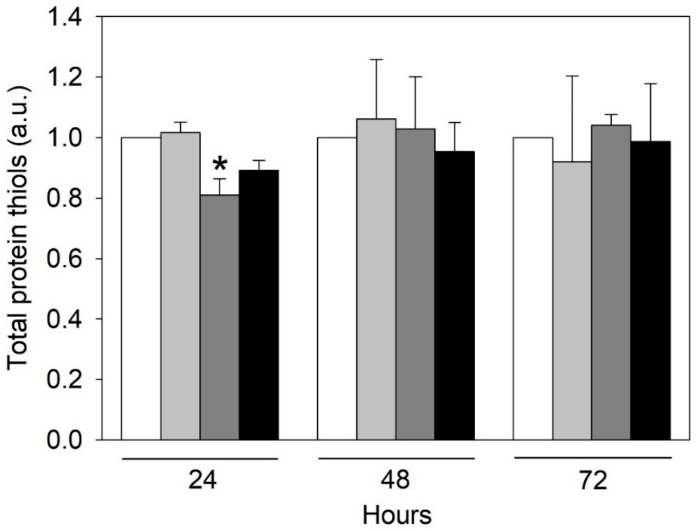
Effect of urea on the total amount of protein thiols in HMEC-1. Histograms showing the protein thiol level measured in control cells and cells treated with 0.25, 2, or 5 g/L urea for 0, 24, 48, or 72 h. Data are expressed as the mean ± SD. * *p* < 0 05.

**Figure 3 ijms-24-00691-f003:**
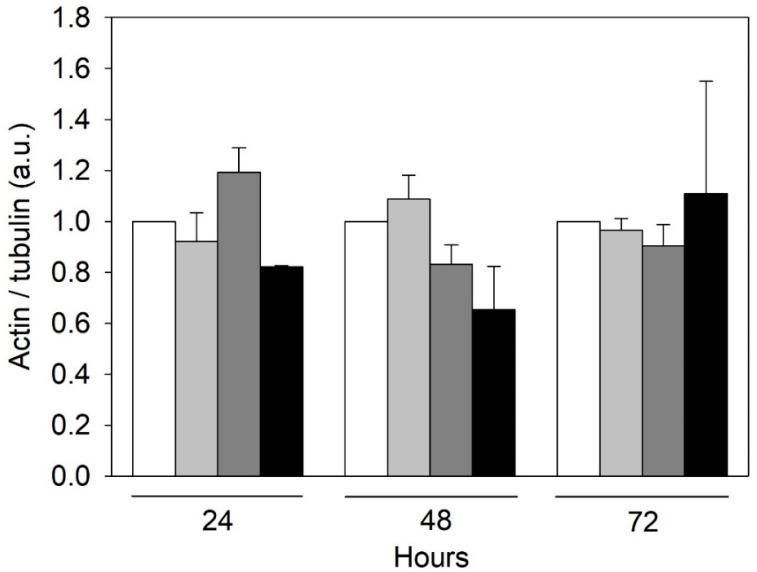
Effect of urea on the actin amount in HMEC-1. Histograms showing the actin level measured in control cells and cells treated with 0.25, 2, or 5 g/L urea for 0, 24, 48, or 72 h. Tubulin was used as a housekeeping protein. Data are expressed as the mean ± SD.

**Figure 4 ijms-24-00691-f004:**
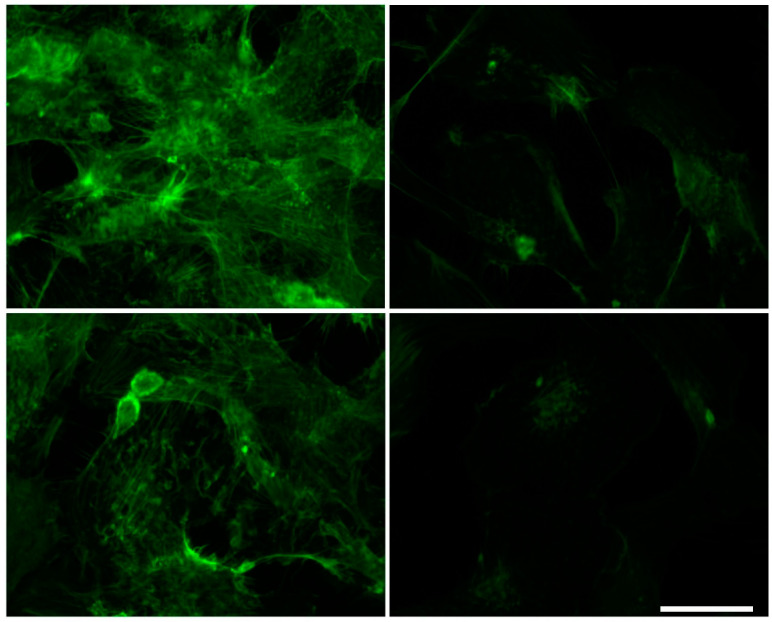
Effect of urea on the microfilament organization in HMEC-1. On the (**left**), control cells at 72 h. On the (**right**), cells treated with 5 g/L urea for 72 h.

**Figure 5 ijms-24-00691-f005:**
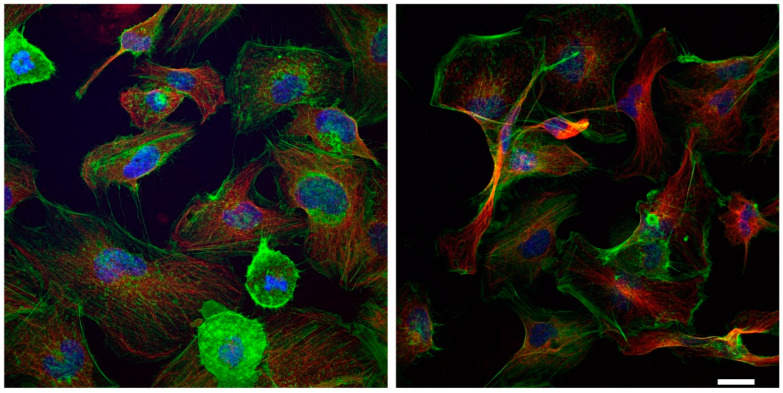
Effect of urea on microfilament and microtubule organization in HMEC-1. On the (**left**), control cells at 72 h. On the (**right**), cells treated with 5 g/L urea for 72 h. Green, microfilaments; red, microtubules; blue, DNA in the nucleus.

**Figure 6 ijms-24-00691-f006:**
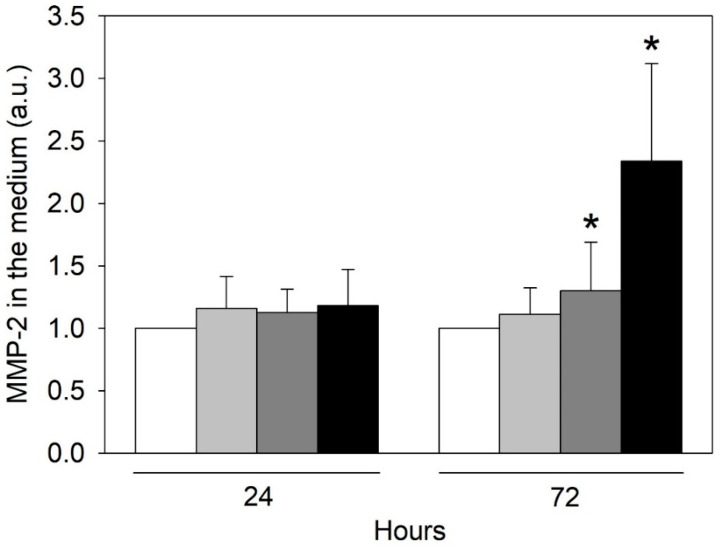
Effect of urea on the release of MMP-2 in the culture medium by HMEC-1. Histograms showing MMP-2 level measured in the culture medium of control cells and cells treated with 0.25, 2, or 5 g/L urea for 0, 24, 48, or 72 h. Data are expressed as the mean ± SD. * *p* < 0 05.

**Figure 7 ijms-24-00691-f007:**
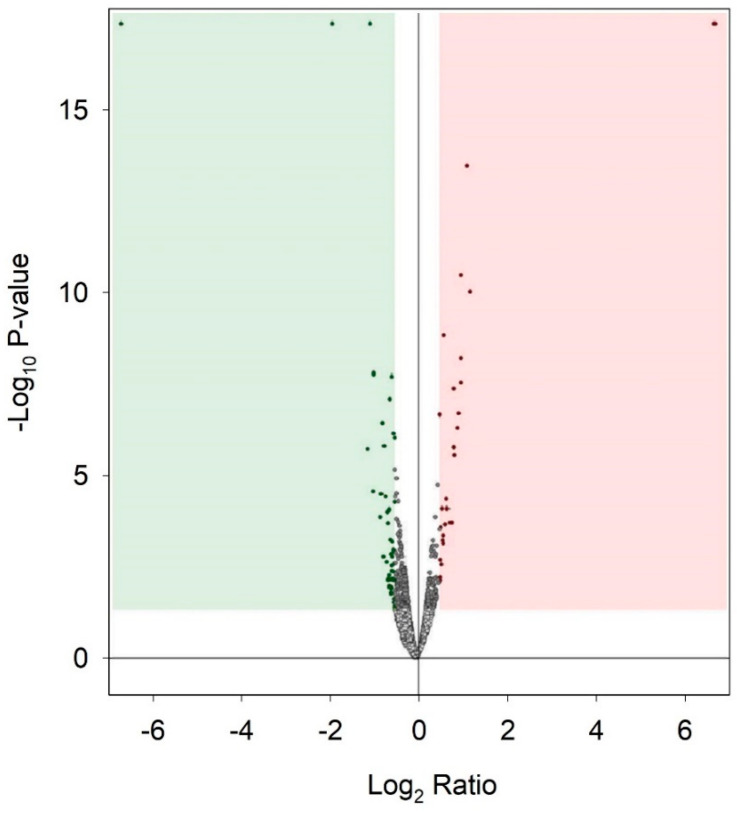
Effect of urea on protein expression in HMEC-1. Volcano plot that compares protein expression of cells treated with 0.25 g/L urea vs. 5 g/L urea for 72 h. Down-regulated proteins are in green. Up-regulated proteins are in red.

**Figure 8 ijms-24-00691-f008:**
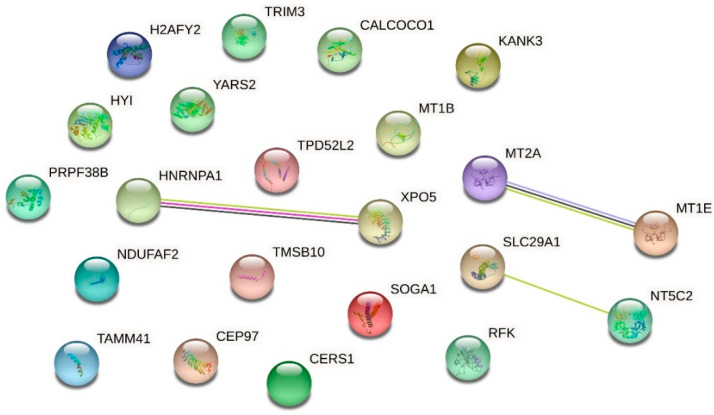
Network of up-regulated proteins in HMEC-1 exposed to urea. The network, obtained with the software STRING, includes proteins that were found to be up-regulated in cells treated with 5 g/L urea compared to cells treated with 0.25 g/L urea for 72 h.

**Figure 9 ijms-24-00691-f009:**
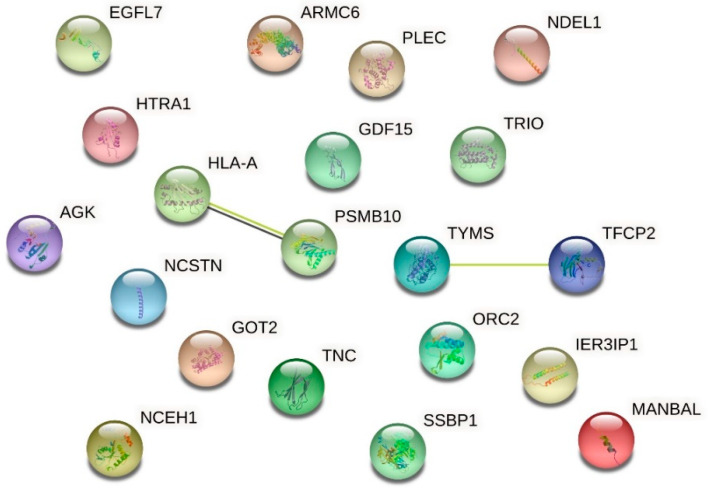
Network of down-regulated proteins in HMEC-1 exposed to urea. The network, obtained with the software STRING, includes proteins that were found to be down-regulated in cells treated with 5 g/L urea compared to cells treated with 0.25 g/L urea, for 72 h.

**Figure 10 ijms-24-00691-f010:**
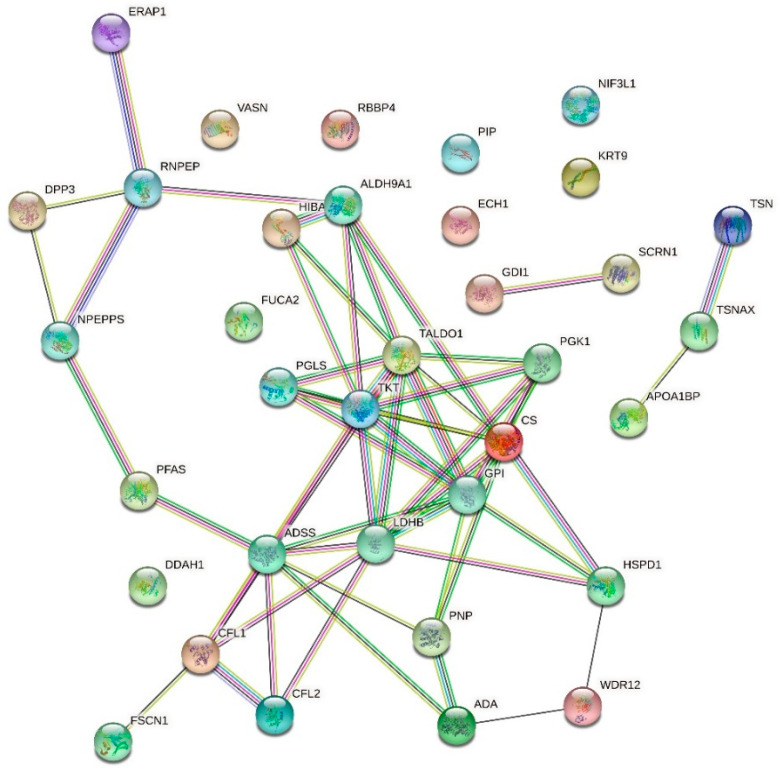
Network of up-regulated proteins in the HMEC-1 secretome. The network, obtained with the software STRING, includes the secreted proteins that were found to be up-regulated in cells treated with 5 g/L urea for 72 h compared to control cells.

**Figure 11 ijms-24-00691-f011:**
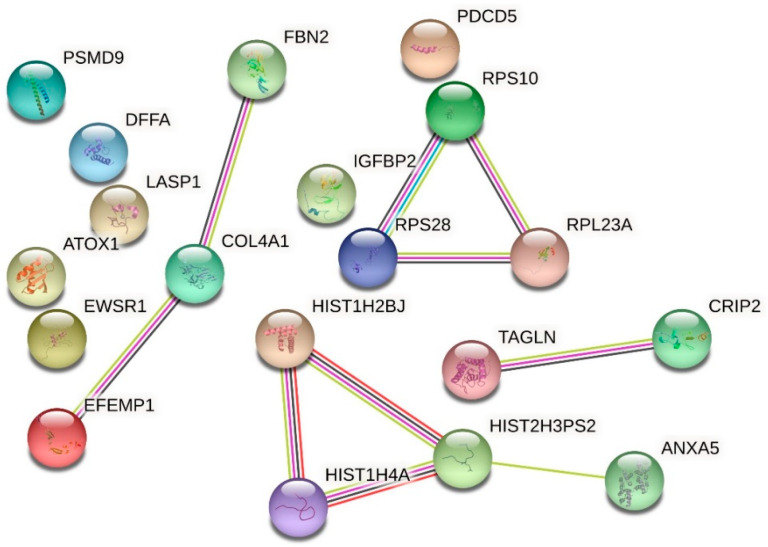
Network of down-regulated proteins in the HMEC-1 secretome. The network, obtained with the software STRING, includes the secreted proteins that were found to be down-regulated in cells treated with 5 g/L urea for 72 h compared to control cells.

**Figure 12 ijms-24-00691-f012:**
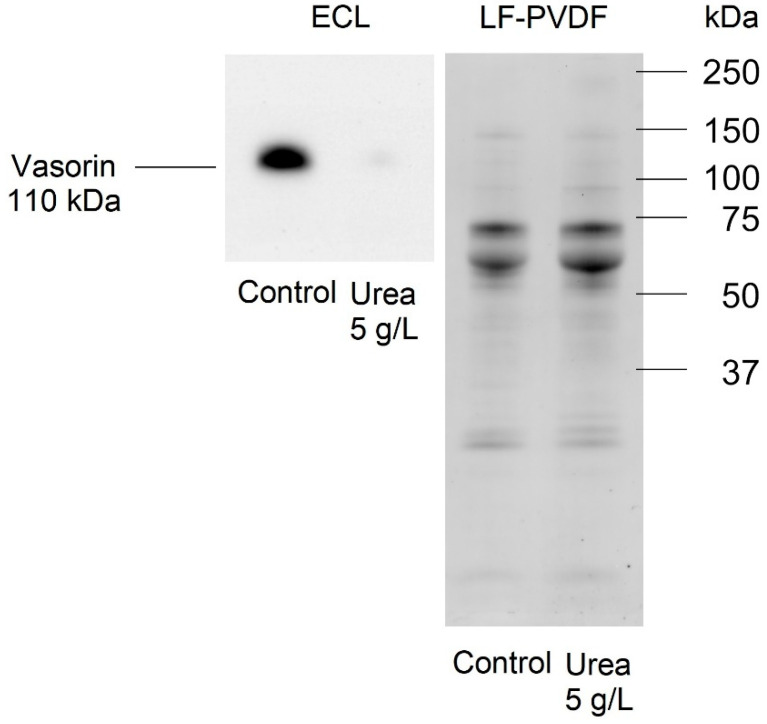
Effect of urea on the expression of the secreted form of vasorin. ECL and LF-PVDF membrane stained with a total protein stain to check the transfer efficiency showing vasorin level assessed by Western blot in the secretome of control HMEC-1 and HMEC-1 treated with 5 g/L urea for 72 h.

**Table 1 ijms-24-00691-t001:** Differentially expressed (modulated) intracellular proteins following exposure of HMEC-1 to urea.

Up-Regulated Proteins				
Accession	Description	Abundance Ratio		*p*-Value
		**5 g/L vs. 0.25 g/L**		
**O94964**	Isoform 2 of Protein SOGA1	2.267		0.000000020868
**Q5VTL8**	Pre-mRNA-splicing factor 38B	2.156		0.000000000010
**Q8IW35**	Centrosomal protein of 97 kDa	1.953		0.000004938470
**P09651**	Isoform A1-A of Heterogeneous nuclear ribonucleoprotein A1	1.949		0.000001279087
**Q9HAV4**	Exportin-5	1.944		0.000000008737
**P04732**	Metallothionein-1E	1.865		0.000033970136
**P07438**	Metallothionein-1B	1.865		0.000033970136
**Q9P0M6**	Core histone macro-H2A.2	1.830		0.000073230796
**Q9Y2Z4**	Tyrosine--tRNA ligase, mitochondrial	1.739		0.000365094872
**Q8N183**	NADH dehydrogenase [ubiquinone] 1 alpha subcomplex factor 2	1.722		0.000007391038
**P49902**	Cytosolic purine 5′-nucleotidase	1.719		0.000237839101
**Q6NY19**	KN motif and ankyrin repeat domain-containing protein 3	1.669		0.013682890309
**O43399**	Isoform 3 of Tumor protein D54	1.594		0.006287115623
**Q5T013**	Putative hydroxypyruvate isomerase	1.584		0.003957519742
**Q9P1Z2**	Calcium-binding and coiled-coil domain-containing protein 1	1.547		0.014584529276
**P63313**	Thymosin beta-10	1.508		0.000000312149
**Q969G6**	Riboflavin kinase	1.506		0.025817204876
**O75382**	Tripartite motif-containing protein 3	1.499		0.036583959883
**Q99808**	Equilibrative nucleoside transporter 1	1.489		0.032611308850
**Q96BW9**	Phosphatidate cytidylyltransferase, mitochondrial	1.464		0.006245305736
**P27544**	Ceramide synthase 1	1.445		0.017775493602
**P02795**	Metallothionein-2	1.418		0.000036164289
**Down-regulated proteins**				
**Accession**	**Description**	**Abundance Ratio**		***p*-Value**
		**5 g/L vs. 0.25 g/L**	**0.25 g/L vs. 5 g/L**	
**Q12800**	Alpha-globin transcription factor CP2	0.265	3.773	0.0000000000000015
**O75962**	Triple functional domain protein	0.454	2.202	0.0002507027368594
**Q04837**	Single-stranded DNA-binding protein, mitochondrial	0.474	2.109	0.0000000000000015
**Q9NQG1**	Protein MANBAL	0.495	2.020	0.0027647462862580
**Q13416**	Origin recognition complex subunit 2	0.503	1.988	0.0000031299038550
**P04818**	Thymidylate synthase	0.558	1.792	0.0101710429818778
**Q92542**	Nicastrin	0.564	1.773	0.0031084727012270
**Q9GZM8**	Nuclear distribution protein nudE-like 1	0.575	1.739	0.0000538096333800
**P24821**	Isoform 4 of Tenascin	0.593	1.686	0.0002257365561964
**Q9Y5U9**	Immediate early response 3-interacting protein 1	0.606	1.650	0.0035816066338457
**P01892**	HLA class I histocompatibility antigen, A-2 alpha chain	0.622	1.607	0.0074612565194133
**Q53H12**	Acylglycerol kinase, mitochondrial	0.629	1.589	0.0136828903087287
**P40306**	Proteasome subunit beta type-10	0.644	1.552	0.0064794736331468
**Q99988**	Growth/differentiation factor 15	0.649	1.540	0.0000143152780648
**Q9UHF1**	Epidermal growth factor-like protein 7	0.654	1.529	0.0308946734033236
**P24821**	Tenascin	0.666	1.501	0.0000035594104464
**Q6NXE6**	Armadillo repeat-containing protein 6	0.674	1.483	0.0340911967228681
**Q15149**	Isoform 3 of Plectin	0.691	1.447	0.0496321589542498
**Q92743**	Serine protease HTRA1	0.693	1.443	0.0001037941772726
**P00505**	Aspartate aminotransferase, mitochondrial	0.700	1.428	0.0001349131929805
**Q6PIU2**	Neutral cholesterol ester hydrolase 1	0.707	1.414	0.0045281605174441

**Table 2 ijms-24-00691-t002:** Differentially expressed (modulated) secreted proteins following exposure of HMEC-1 to urea.

Up-Regulated Proteins				
Accession	Description		Abundance Ratio	*p*-Value
			**Control/Urea (5 g/L)**	
**APOA1BP**	NAD(P)H-hydrate epimerase		49.966	0.000000000000022
**O94760**	N(G),N(G)-dimethylarginine dimethylaminohydrolase 1		45.754	0.000000000000091
**A0A024R3X4**	Epididymis secretory sperm binding protein		30.065	0.000000000046539
**P30520**	Adenylosuccinate synthetase isozyme 2		26.540	0.000000000936884
**A0A024R3V7**	NIF3-like protein 1		14.996	0.000009726416165
**A0A024RB75**	Citrate synthase		14.257	0.000000573618455
**P12273**	Prolactin-inducible protein		14.216	0.000000591547361
**P55786**	Puromycin-sensitive aminopeptidase		14.108	0.000000641574461
**Q549N0**	Cofilin 2 (Muscle), isoform CRA_a		12.992	0.000001555188153
**Q8N7G1**	Purine nucleoside phosphorylase		12.660	0.000001995520621
**P29401**	Transketolase		11.977	0.000003566403024
**Q16658**	Fascin		11.654	0.000004723432374
**Q9BTY2**	Plasma alpha-L-fucosidase		11.477	0.000018168658036
**Q13011**	Delta(3,5)-Delta(2,4)-dienoyl-CoA isomerase, mitochondrial		11.337	0.000007886470783
**PFAS**	Phosphoribosylformylglycinamidine synthase, isoform CRA_b		11.234	0.000006831656478
**Q53T99**	Ribosome biogenesis protein WDR12		9.173	0.000048900585780
**Q6EMK4**	Vasorin		9.026	0.000641197347120
**V9HWF4**	Phosphoglycerate kinase		8.109	0.000151428360979
**G3V180**	Dipeptidyl peptidase 3		7.275	0.002621057250876
**GDI1**	Rab GDP dissociation inhibitor		6.991	0.000533863830113
**P49189**	4-trimethylaminobutyraldehyde dehydrogenase		6.658	0.000770649443919
**P00813**	Adenosine deaminase		6.281	0.001233369407633
**P31937**	3-hydroxyisobutyrate dehydrogenase, mitochondrial		5.937	0.008777288491631
**E9PK25**	Cofilin-1		5.435	0.003602441604062
**SCRN1**	Secernin 1		5.201	0.020151555976657
**GPI**	Glucose-6-phosphate isomerase		5.044	0.006722750720656
**P37837**	Transaldolase		5.013	0.006387658270684
**P07195**	L-lactate dehydrogenase B chain		4.842	0.008086580808851
**A0A024R3V8**	Translin-associated factor X, isoform CRA_c		4.789	0.030367926630212
**Q09028**	Histone-binding protein RBBP4		4.762	0.021788773185764
**HEL-S-304**	6-phosphogluconolactonase		4.740	0.027747024650690
**Q15631**	Translin		4.612	0.011041331019512
**RNPEP**	Aminopeptidase B		4.580	0.030367926630212
**Q9NZ08**	Endoplasmic reticulum aminopeptidase 1		3.997	0.025908828760954
**P35527**	Keratin, type I cytoskeletal 9		3.948	0.027944488189618
**Down-regulated proteins**				
**Accession**	**Description**	**Abundance Ratio**		***p*-Value:**
		**Control/Urea (5 g/L)**	**Urea (5g/L)/Control**	
**P35556**	Fibrillin-2	0.068	14.705	0.000001550
**P06899**	Histone H2B type 1-J	0.082	12.195	0.000010319
**P02462**	Collagen alpha-1(IV) chain	0.127	7.874	0.002994145
**P08758**	Annexin A5	0.127	7.874	0.000533864
**Q5TEC6**	Histone H3	0.128	7.812	0.000563535
**P18065**	Insulin-like growth factor-binding protein 2	0.129	7.751	0.000601115
**Q01844**	RNA-binding protein EWS	0.131	7.633	0.005687063
**P46783**	40S ribosomal protein S10	0.136	7.352	0.000918185
**P62857**	40S ribosomal protein S28	0.157	6.369	0.009562979
**O00273**	DNA fragmentation factor subunit alpha	0.164	6.097	0.020667882
**B2R4R0**	Histone H4	0.171	5.847	0.005017696
**O14737**	Programmed cell death protein 5	0.182	5.494	0.033480893
**EFEMP1**	EGF containing fibulin-like extracellular matrix protein 1 isoform 1	0.189	5.291	0.010064879
**P52943**	Cysteine-rich protein 2	0.189	5.291	0.044393972
**Q01995**	Transgelin	0.208	4.807	0.017871069
**O00233**	26S proteasome non-ATPase regulatory subunit 9	0.209	4.784	0.042721853
**Q14847**	LIM and SH3 domain protein 1	0.210	4.761	0.019123177
**E5RIM7**	Copper transport protein ATOX1	0.218	4.587	0.023827383
**P62750**	60S ribosomal protein L23a	0.231	4.329	0.032476814

## Data Availability

Data are included within the article.

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
