# Peer review of "Effects of Physiological and Pathological Urea Concentrations on Human Microvascular Endothelial Cells"

_ijms, 2022, doi:10.3390/ijms24010691_

Round 1
Reviewer 1 Report
The authors demonstrated the effect of urea on the human microvascular endothelial cells with clinical relevant concentration. From the in vitro studies, the urea influence the proliferation ability marginally at 72 hours.The MMP-2 increased in urea exposed HMEC cells. From the bioinformatics, the secretome interaction, the authors tried to discuss the protein asssociated with MMP and the secretome vasorin.
The authors provided the marginal effect of urea on the HMEC with MMP, and the proteomics of different concentration of urea and secretome between pathologic concentration of urea and control. Finally, the expression of vasorin was performed. Large amount of information was provided without main idea.
1. The main methods used in the study was the bioinformatics such as proteomics and secretomes. However, low information was mentioned in the introduction section, especially the secretomes. Besides, the introduction from line 45 to 71 could be concised.
2. Result:
2.2: The result did not fully support that the oxidative stress increased in HMEC, since the protein thiol only increased at 24 hours at 2g.
2.7 From the secretome, the expression of vasorin was not the highest in the up-regulated protein categories. The authors should add the reason for choosing vasorin other than other proteins. Besides, the description between line 209-219 should be concised.
3. discussion:
line 230-256: I suggested the authors to concise the section since the knowledge of urea is commonly understand by the readers or even the medical students. The authors should focus on the mechanism on endothelial injury.
Line 273-324: The increase in MMP2 and higher apoptosis at pathologic concentration was consistent with previous studies. I suggest the authors to concise the literature to the articles consistent with the authors result only because the authors novel finding was mainly on the secretome.
Line 341-390:
My main concern was : since annexin 5, keratin or fibirllin 2 all increased in urea treated cells, why authors only detect the amount of vasorin?
I suggest the authors to address the logics.
Overall, the introduction and discussion were too redundant. Thorough revision is needed.
Reviewer 2 Report
The article is devoted to an actual topic. The introduction describes in detail the practical significance. Uremic toxins are powerful stimulants that affect all cells. The conclusions drawn during the study are logical and flow smoothly from the conclusion.
The influence of urea on the proliferative potential of endothelial cells is the basis for the development of socially significant human diseases. The paper presents the main mechanisms of endothelial cell proliferation under the influence of urea from a modern perspective, and identifies the main targets.
The strengths of the article are a thorough fundamental approach used to study the mechanisms of influence on endothelial cells. However, there are weaknesses, this is the lack of clinical verification. Detailed protein interactions have been constructed and indicated, but they have not been verified. There is no data on studies on living objects, models of diseases of the cardiovascular system.
To improve the presentation of the article, it is necessary to supplement the conclusion section, in which it is necessary to add the practical significance of the work for the correction of pathological human conditions.
Round 2
Reviewer 1 Report
The authors have made the relevant revision.
I have no further comments.